# Metal coordinating inhibitors of Rift Valley fever virus replication

**Elizabeth Geerling[1], Valerie Murphy[1], Maria C. Mai[1], E. Taylor Stone[1], Andreu Gazquez Casals[1], Mariah Hassert[1], Austin T. O'Dea[1], Feng Cao[2], Maureen J. Donlin[3], Mohamed Elagawany[4], Bahaa Elgendy[4,5], Vasiliki Pardali[6], Erofili Giannakopoulou[6], Grigoris Zoidis[6], Daniel V. Schiavone[7], Alex J. Berkowitz[7], Nana B. Agyemang[7], Ryan P. Murelli[7], John E. Tavis[1], Amelia K. Pinto[1], James D. Brien[1]***

1 Department of Molecular Microbiology and Immunology, Saint Louis University School of Medicine, Saint Louis, Missouri, United States of America, 2 John Cochran Division, Department of Veterans Affairs Medical Center, Saint Louis, Missouri, United States of America, 3 Department of Biochemistry and Molecular Biology, Saint Louis University School of Medicine, Saint Louis, Missouri, United States of America, 4 Center for Clinical Pharmacology, Washington University School of Medicine and University of Health Sciences and Pharmacy, Saint Louis, Missouri, United States of America, 5 Department of Pharmaceutical and Administrative Sciences, University of Health Sciences and Pharmacy, Saint Louis, Missouri, United States of America, 6 Division of Pharmaceutical Chemistry, Department of Pharmacy, School of Health Sciences, National and Kapodistrian University of Athens, Athens, Greece, 7 Department of Chemistry and The Graduate Center of The City University of New York, Brooklyn College, The City University of New York, Brooklyn, New York, United States of America

* James.Brien@health.slu.edu

**Data Availability Statement:** All relevant data are within the paper and its Supporting information files.

**Funding:** This work was funded by the USA National Institutes of Health [grants R01

## Abstract

Rift Valley fever virus (RVFV) is a veterinary and human pathogen and is an agent of bioterrorism concern. Currently, RVFV treatment is limited to supportive care, so new drugs to control RVFV infection are urgently needed. RVFV is a member of the order *Bunyavirales*, whose replication depends on the enzymatic activity of the viral L protein. Screening for RVFV inhibitors among compounds with divalent cation-coordinating motifs similar to known viral nuclease inhibitors identified 47 novel RVFV inhibitors with selective indexes from 1.1–103 and 50% effective concentrations of 1.2–56 μM in Vero cells, primarily α-Hydroxytropolones and N-Hydroxypyridinediones. Inhibitor activity and selective index was validated in the human cell line A549. To evaluate specificity, select compounds were tested against a second Bunyavirus, La Crosse Virus (LACV), and the flavivirus Zika (ZIKV). These data indicate that the α-Hydroxytropolone and N-Hydroxypyridinedione chemotypes should be investigated in the future to determine their mechanism(s) of action allowing further development as therapeutics for RVFV and LACV, and these chemotypes should be evaluated for activity against related pathogens, including Hantaan virus, severe fever with thrombocytopenia syndrome virus, Crimean-Congo hemorrhagic fever virus.

## 1. Introduction

*Bunyavirales* is a large order of enveloped viruses with a segmented, negative-polarity, single-stranded RNA genome and includes multiple members that pose a significant risk to public

AI1222669, R01 AI148264, NIH R21 AI124672 to JT, and SC GM111158 to RM], the Special Account for Research Grants from the National and Kapodistrian University of Athens [grant 15725] to GZ, and a seed grants from the School of Medicine and the Department of Molecular Microbiology and Immunology, Saint Louis University School of Medicine to JT, AKP and JDB, McNair Scholars Program to MCM. The funders had no role in study design, data collection and analysis, decision to publish, or preparation of the manuscript.

health: Rift Valley fever virus (RVFV), La Crosse virus (LACV), Hantaan virus, severe fever with thrombocytopenia virus (SFTSV), and Crimean-Congo hemorrhagic fever virus (CCHFV) [1, 2]. RVFV is an arbovirus transmitted to animals by mosquito vectors. It is traditionally endemic in eastern and southern Africa but has recently expanded its range throughout sub-Saharan Africa and parts of the Middle East. RVFV is a serious veterinary pathogen, causing Rift Valley fever in domestic animals including cattle, horses, sheep, goats, and camels. Rift Valley fever is characterized by fever, hemorrhage, diarrhea, death, and nearly complete spontaneous abortions in infected animals. Veterinary outbreaks of RVFV infections can reach epidemic proportions, particularly in rainy years [3, 4]. Often, RVFV has a severe economic impact in regions with many affected herds.

Humans can also be infected by RVFV via contact with infected animal body fluids or tissues, by breathing droplets contaminated with RVFV, or less frequently via mosquito bites; with human to human transmission of RVFV being rare [2]. Most human infections are either asymptomatic or cause mild fever with hepatic involvement. However, 8–10% of infections become severe, where symptoms can include lesions to the eye which cause blindness in 50% of ocular cases, encephalitis, gastrointestinal dysfunction, jaundice, joint/muscle pain, hemorrhagic fever, disorientation/hallucination, and partial paralysis ([5], Reviewed in [2]). Hemorrhagic fever is rare (~1% of cases) but has a ~50% fatality rate in cases where it occurs. Human RVFV infections can be diagnosed by ELISA or RT-PCR assays, but treatment is limited to supportive care [2]. Currently, there is a live-attenuated veterinary vaccine approved for RVFV (MP12), which has undergone a phase I clinical trial for use in humans (NCT00415051).

In addition to RVFV, LACV is an arbovirus found throughout the midwestern United States, with childhood infections commonly underdiagnosed due to a lack of available diagnostics and therapeutics [6, 7]. Currently there are 50–150 cases of neuroinvasive disease reported annually [8]. Since 2011, LACV has continued to spread beyond the Midwest and into northeastern, mid-Atlantic and southern states, resulting in 700 cases of neuroinvasive disease since 2011 [9].

All viruses of the order *Bunyavirales* replicate within the cytoplasm, facilitated by the L protein which is comprised of the RNA-dependent RNA polymerase (RdRp), cap binding domain and viral endonuclease. These activities can make Bunyavirales a target for both direct acting antivirals as well as host direct antivirals [10–12]. The virally encoded enzymes requires either $Mg^{++}$ or $Mn^{++}$ ions for catalysis and hence viral replication [1, 13]. This is analogous to the $Mg^{++}$-binding motif of the Influenza Virus PA cap-snatching endonuclease [14] and bears significant similarity to the D..E..D..D or D..D..E motifs found in ribonucleases H and viral integrases [15]. Inhibiting viral enzymatic function by altering $Mg^{++}$ or $Mn^{++}$ ions can block viral replication, as has been shown for the HIV integrase [16], the HIV ribonuclease H [17], the Hepatitis B Virus (HBV) ribonuclease H [18], and the influenza virus cap-snatching enzyme [19, 20]. The most common mode of inhibition is for small molecules to chelate the $Mg^{++}$ ions in viral enzymatic active sites, with specificity and affinity modulated by additional contacts between the inhibitors and the enzymes [21–23], and sometimes also by contacts with the nucleic acid substrate [24]. This metal-chelating mechanism is used by the HIV integrase inhibitors Bictegravir [25], Dolutegravir [26, 27], Elvitegravir [28], and Raltegravir [28], and the Influenza Virus PA cap-snatching inhibitor Baloxavir marboxil [29]. The US Food and Drug Administration has approved 62 drugs that act by coordinating active-site cations in metalloenzymes as of 2017 [30], making active site metal ion chelation a well-established drug mechanism.

In these studies, we hypothesized that metal chelating compounds similar to inhibitors of the HIV and HBV ribonucleases H, the HIV integrase, and the Influenza Virus PA endonuclease would inhibit RVFV and LACV replication. This hypothesis is based on i) the inhibitory

mechanism employed by metal chelating compounds against metalloenzymes, ii) the essential nature of the L protein enzymatic function for viral replication [31], iii) the structural similarities of $Mg^{++}$-dependent viral endoribonucleases, even between the *Phenuiviridae* and *Peribunyaviridae* families [15, 31, 32], and iv) the successes in developing drugs for HIV and Influenza virus that act by chelating the catalytic $Mg^{++}$ ions. In order to evaluate metal chelating compounds, we developed and validated assays to quantify live virus growth in the presence of potential antiviral compounds. Using these assays, we screened 397 compounds to identify chemotypes with antiviral potential in both non-human primate and human cell lines. We then further identified compounds which had activity against the *Phlebovirus* RVFV and the closely related *Orthobunyavirus*, LACV. To determine specificity against bunyaviruses versus RNA viruses, we measured the *in vitro* efficacy of the best compounds against the positive strand RNA flavivirus, Zika. Additionally, we developed live virus assays to quantify antiviral compound activity validated these assays using known antivirals.

## 2. Materials and methods

### 2.1. Compound acquisition and synthesis

Commercially acquired compounds are indicated by the vendor's name and catalog number in S1 Table. Thiotropolones (TTP) compounds were synthesized as described in [33]. α-Hydroxytropolones (αHT) compounds were synthesized as described in [34]. For 265, 308, and 311 see [35]. For 169 and 362 see [36]. For 385, 694, 696, 698, 700, 702, 704, 703, 838, and 840 see [37]. For 321, 336, 358, and 359 see [38]. For 388, 389, 390 and 539 see [39]. For 330 and 331 see [40]. For 113, 118 and 120 see [41]. For 260 see [42]. For 111 see [34]. For 335 see [43].

The novel N-hydroxypyridinediones (HPD) compounds were synthesized following a three-step synthetic procedure (S1 Data, Scheme 1). The key structure 5-acetyl-1-(benzyloxy)-6-hydroxy-4-methylpyridin-2(1*H*)-one (ZEV1) was synthesized with an improved yield of 75% by refluxing a mixture of *O*-benzyl hydroxylamine (1 eq) and diketene (2 eq) in the presence of triethylamine (1 eq) in dry toluene. Subsequently, the benzyl group was cleaved by catalytic hydrogenation over 10% palladium on carbon to afford the target compound ZEV2 almost quantitatively. 5-Acetyl-1,6-dihydroxy-4-methylpyridin-2(1*H*)-one (ZEV2) was coupled with the appropriate substituted aniline using sulfuric acid as catalyst in absolute ethanol at reflux. The desired compounds were obtained in good yields ranging from 60% to 70%, with the only exception being 668 (ZEV-V5) which was isolated in an overall yield of 25%. HPD compounds not published previously include 515–518, 668 and 670; preparation procedures and characterization data of compounds are in S1 Data. Nucleoside analogues ribavirin and β-D-N4-Hydroxycytidine N4-Hydroxycytidine were purchased from Cayman Chemical. Compounds were diluted to 10 mM in DMSO and stored in single-use aliquots in opaque tubes at -20 ˚C.

### 2.2. Cells and viruses

RVFV strains MP-12 and ZH501, as well as LACV strain original (BEI NR-540) were passaged in Vero E6 cells (ATCC® CRL-1586™) before clarification by centrifugation at 3,000 rpm for 30 minutes and stored at -80˚C until further use. RVFV isolates were a kind gift of Drs. M. Buller (Saint Louis University) and A. Hise (Case Western Reserve University). MP-12 is a BSL-2 vaccine strain of RVFV and is not classified as a select agent allowing easier assay development. The ZIKV strain, PRVABC59 was a kind gift of Robert Lanciotti (CDC). Virus inhibition assays and toxicity assays, described below, were completed in both Vero E6 and A549 cells (ATCC CCL-185). Unless otherwise specified, all cells were cultured in Dulbecco's Modified

Eagle Medium (Sigma- D5796-500ML) containing 1% HEPES (Sigma- H3537-100ML) and 5% FBS (Sigma- F0926) at 37˚C, 5% $CO_2$. Studies with infectious RVFV-ZH501 viruses were approved by the SLU IBC and were conducted in our select agent registered A/BSL-3 laboratory.

## 2.3. Focus Forming Assay (FFA)

FFAs are used to quantify infectious virus and are the basis for the antiviral compound inhibition assay. Briefly, 100µL of Vero E6 cells at a concentration of $3x10^5$ cells/ml were plated in a 96-well flat bottom plate resulting in a confluency of 90–95% the day prior to the assay. To quantify viral stocks, ten-fold serial dilutions of virus supernatants were then made in a 96-well round bottom plate containing 5% DMEM media before being added to the Vero cell monolayer and allowed to adsorb for one hour in an incubator with 37˚C, 5% $CO_2$. Following virus adsorption, a solution of 2% methylcellulose (Sigma-M0512-250G) was diluted 1:1 in 5% DMEM and warmed to room temperature. The methylcellulose-media mixture overlay was added to the plate by adding 125 µL of overlay media to each well and returned to an incubator with 37˚C, 5% $CO_2$ for 24 hours. Plates were then fixed in a solution of 5% paraformaldehyde (PFA) diluted in tissue culture grade 1X PBS, then washed in 1X PBS for 15 minutes. Foci were visualized by an immunostaining protocol using anti-nucleocapsid protein antibody (1D8) diluted 1:5000 to detect RVFV and anti-Gc protein antibody (4C12A1) diluted 1:5000 to detect LACV with FFA staining buffer (1X PBS, 1mg/ml saponin (Sigma: 47036)) as a primary detection antibody overnight at 4˚C. The anti-nucleocapsid protein antibody (1D8) was obtained from Joel Dalrymple and Clarence J Peters (USAMRIID) via BEI resources. The secondary antibody consisted of goat anti-mouse conjugated horseradish peroxidase (Sigma: A-7289) diluted 1:5,000 in FFA staining buffer and allowed to incubate for 2 hours at room temperature. Foci were visualized using KPL TrueBlue HRP substrate and allowed to develop for 10–15 minutes, or until blue foci are visible. The reaction was then quenched by washing with Millipore water. RVFV ZH501 foci assays were measured within the BSL3 facility, while RVFV MP12 and LACV viral foci were quantified with an automated ELISPOT machine (CTL universal S6) using the Immunospot software suite.

## 2.4. Antiviral compound efficacy assay

For RVFV (MP-12 and ZH501), Vero cells were plated at $3x10^5$ cells per mL in a flat bottom 96 well plate. Twenty-four hours later they were infected with RVFV at a multiplicity of infection of 0.005. Compound was added and plates were incubated for 1 hour at 37˚C and then cells were overlayed with methylcellulose. After 24 hours plates were fixed, RVFV infectious foci were stained and quantified as described for the FFA above. Data is normalized to PBS, using the average number of foci in each individual assay and presented as Focus Forming Units (FFU). All assays required the range of foci to be 70–90 foci per control well for the assay to pass quality control.

For RVFV (MP-12)- A549 cells, a human lung epithelial cell line, were plated at $3x10^5$ cells per mL in 96 well plates and incubated at 37˚C for 24 hours. Cells were then infected with RVFV strain MP-12 at a multiplicity of infection of 0.005. Plates were incubated for 1 hour at 37˚C and then cells were overlayed with methylcellulose. After 24 hours plates were fixed, RVFV infectious foci were stained and quantified as described for the FFA above.

For LACV (original), Vero cells were plated at $2x10^5$ cells per mL in a flat bottom 96 well plate. Twenty-four hours later they were infected with LACV at a multiplicity of infection of 0.01. Compound was added and plates were incubated for 1 hour at 37˚C and then cells were overlayed with methylcellulose. After 24 hours plates were fixed, LACV infectious foci were

stained using the murine anti-LACV Gc antibody, detected with an anti-mouse HRP secondary antibody and quantified as described for the FFA above.

For Zika virus (ZIKV) we have used our previously validated assay [44, 45], briefly Vero cells were plated at $4x10^5$ cells per mL in a flat bottom 96 well plate. Twenty-four hours later they were infected with ZIKV at a multiplicity of infection of 0.01. Compound was added and plates were incubated for 1 hour at 37°C and then cells were overlayed with methylcellulose. After 48 hours plates were fixed, ZIKV infectious foci were stained using the ZIKV cross-reactive antibody (4G2), detected with an anti-mouse HRP secondary antibody and quantified as described for the FFA above.

Fifty percent effective concentrations ($EC_{50}$) for key hits were determined by screening for suppression of viral growth using an eight point, 2.5-fold dilution series of the compounds starting at 100 μM. $EC_{50}$ values were calculated by non-linear curve fitting in GraphPad Prism v8.

## 2.5. Compound cytotoxicity

Initial compound cytotoxicity was estimated in Vero cells in the primary antiviral compound experiments by staining the virally infected, compound treated cells with crystal violet after viral foci had been quantified. After cells were stained with crystal violet wells were washed 2x with water. The crystal violet dye was then extracted using 50% ethanol and absorbance was measured at 570nm in an ELISA plate reader. Estimated 50% cytotoxic concentration ($CC_{50}$) values were derived by non-linear curve fitting in GraphPad Prism of the four data points derived from the primary screen.

Quantitative $CC_{50}$ values were measured in two systems. Compound cytotoxicity in uninfected A549 cells was determined using the CytoTox-GloTM Cytotoxicity Assay (Promega) according to the manufacturer's instructions. Briefly, A549s were seeded at $3x10^5$ cells per mL in 96 well plates and incubated for 24 hours at 37°C. An eight point, 3-fold dilution series of compounds was added to the cellular monolayer starting with the highest concentration, 600 μM, in addition to DMSO and PBS as controls. Plates were incubated for 48 hours at 37°C, then the AAF-Glo reagent was added to the cells for 15 minutes and luminescence was measured to determine the number of dead cells. Lysis buffer was added next for 15 minutes, then luminescence was measured to determine the total cell number. The dead cell number was then subtracted from the total cell number to generate the viable cell number.

Second, cytotoxicity was measured in the HepG2-derived hepatoblastoma cell line, HepDES19 [46]. Cells were treated with a range of compound concentrations in a final DMSO concentration of 1% for three days and mitochondrial function was measured by MTS assays as described [47]. $CC_{50}$ values were then calculated by non-linear curve fitting in GraphPad Prism v8.

## 2.6. RNaseH inhibition reactions

Activity of human RNaseH was measured using a FRET assay in which the RNA:DNA heteroduplex was formed by annealing an 18 nucleotide-long RNA oligonucleotide with a fluorescein label at the 3' end to a complementary DNA oligonucleotide with an Iowa Black quencher at the 5' end. RNaseH activity cleaves the RNA, permitting the fluorescein to diffuse away from the quencher, increasing fluorescence. The oligonucleotides employed were:

DNA: 5'-IABkFQ-AGC TCC CAG GCT CAG ATC-3' (IABkFQ: Iowa Black quencher)

RNA: 5'- GAU CUG AGC CUG GGA GCU FAM-3' (FAM: Fluorescein fluorophore).

Recombinant human RNaseH 1 was purified from *E. coli* as described in [48]. Enzyme and substrate (12.5 nM) were combined in 100 mM NaCl, 50 mM HEPES pH 8.0, 2 U RNase OUT (ThermoFisher), and test compound in a final concentration of 1% DMSO. Reactions were started by adding $MgCl_2$ to 5 mM and incubating at 37°C for 90 min. with detection of fluorescence every 2 min. in a plate reader. The initial rate was determined for each compound concentration, and $IC_{50}$ values were determined from the reaction rates by non-linear curve fitting in GraphPad Prism.

# 3. Results

## 3.1. Development and validation of a live virus bunyavirus antiviral compound efficacy assay

Based on our previous work optimizing focus forming assays (FFAs) for Zika virus (ZIKV) and SARS-CoV-2, [45, 49, 50] we developed an antiviral compound efficacy assay for RVFV and LACV. We chose to develop an FFA because it is a high-throughput cell-based assay that can capture a range of information about viral growth and replication, such as number of infection foci and foci morphology. Virally induced foci data serve as critical information for the development of antiviral compounds. To establish and validate the assay, we identified an appropriate cell line, defined the primary antibodies capable of detecting viral antigen, and determined an optimal cell concentration for seeding the wells of 96 well plates. Previous work from a number of laboratories demonstrated that Vero E6 cells, a non-human primate African green monkey kidney epithelial cell line, are highly sensitive to RVFV infection and widely available [51].

To identify an appropriate primary antibody to detect viral antigen, we stained a serial dilution of virally infected Vero E6 cells. We evaluated 5 total monoclonal antibodies (mAbs), with 2 antibodies recognizing the Gc glycoprotein (4B6, 3D11), 2 antibodies recognizing the Gn glycoprotein (7B6, 3C10), and one antibody recognizing the nucleocapsid protein (1D8) from the Joel M. Dalrymple and Clarence J. Peters (USAMRIID antibody collection) (Fig 1A). The murine anti-flavivirus mAb 4G2 was used as a negative control. The mAb 1D8 had the best signal to noise ratio and was thus used throughout the rest of these studies.

In order for RVFV to form distinct foci and for the assay to have the highest level of sensitivity, it is critical to plate cells at an optimal density. We examined the impact of cell density on foci formation by plating identical dilutions of RVFV virus stocks on 96-well plates seeded with differing numbers of E6 cells ($7.5 \times 10^4$, $1.5 \times 10^5$ or $3 \times 10^5$ cells/mL). At these concentrations, the monolayers were ~70, 80 and 90 percent confluent, respectively, and the same virus dilution resulted in $1 \times 10^3$ FFU/mL of RVFV MP-12. We observed the highest sensitivity when either $1.5 \times 10^5$ or $3 \times 10^5$ cells/mL were seeded in comparison to $7.5 \times 10^4$ cells/mL (Fig 1B). We have previously tested higher cell densities for FFAs measuring flavivirus and coronavirus replication, and we have noted that cell concentrations higher than $3 \times 10^5$ cells/mL results in an overly confluent monolayer with more cells than can adhere to the wells, which can lead to highly variable viral titer information [45].

To validate the assay design, we completed an antiviral compound inhibition assay by plating Vero E6 cells at $3x10^5$ cells/ml in a 96 well flat bottom plate. These wells were infected with sufficient virus to form ~70–80 foci per well, and cells were treated with serial dilutions of ribavirin or β-D-N4-Hydroxycytidine N4-Hydroxycytidine (NHC/EIDD-1931) starting at 100μM as positive controls. Further, additional wells served as negative controls by being treated with 100μM DMSO as a vehicle control or PBS. The ratio of foci forming units (FFU) in comparison to PBS was measured and the effective concentration 50 ($EC_{50}$) was calculated. Ribavirin

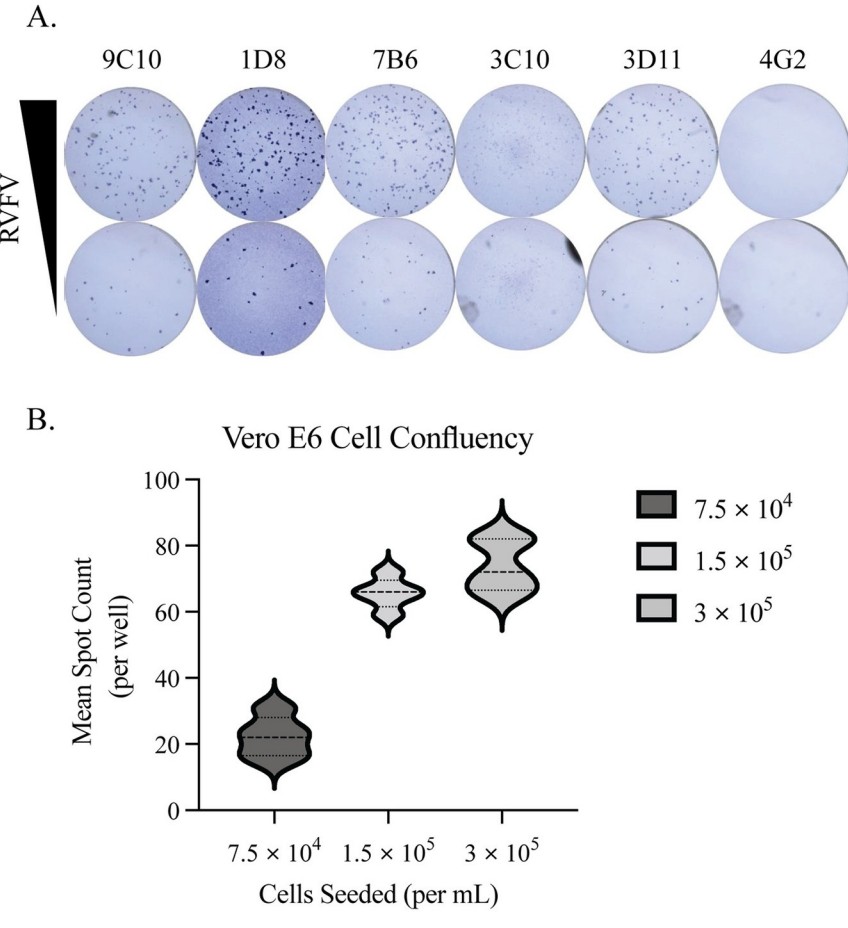

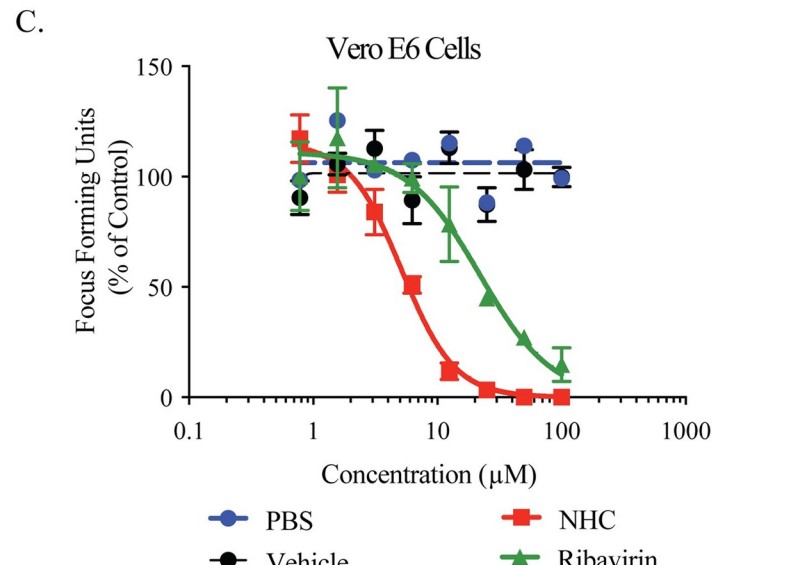

**Fig 1. Development and validation of RVFV antiviral screen. A**. Identification of optimal mAb for the detection of RVFV by mAb staining of a serial dilution of RVFV strain MP-12 in an FFA. **B**. Impact of cell number on the sensitivity of antiviral compound screen. **C**. Evaluation of the sensitivity of the antiviral compound screen based upon the evaluation of ribavirin and β-D-N4-Hydroxycytidine N4-Hydroxycytidine (NHC/EIDD-1931), a known antiviral for RVFV. Data is presented as focus forming units. These data are the cumulation of three independent experiments with technical duplicates.

had an $EC_{50}$ of 22.0 μM, similar to work by other groups [52, 53], while the $EC_{50}$ value for NHC was 5.16 μM (Fig 1C).

## 3.2. Primary screening for antiviral activity

Primary screens were conducted that evaluated 397 compounds either with known metal-chelating motifs or motifs similar to metal-chelating ones. The most common chemotype among the compounds screened was the troponoids (tropones, tropolones (TRP), thiotropolones (TTP), and α-hydroxytropolones (αHTs)), but the compound set also included a wide range of other chemotypes such as the N-hydroxypyridinediones (HPD), flavonoids, N-hydroxy-napthyridinones, dihydronapthalenes (DHN), dioxobutanoic acids, hydroxyxanthanones, thienopyrimidinones, pyridinepiperazinthieonpyrimidins, N-biphenyltrihydroxybenzamides, and aminocyanothiophenes. Almost half of the compounds screened were αHTs as the library from which they were drawn was assembled in support of anti-HBV ribonuclease H antiviral development and the αHTs are a leading chemotype in that effort [38, 54, 55].

To screen these compounds for antiviral activity, FFAs were used. Briefly, cells were infected with RVFV MP-12, treated with 60, 20, 6.7, or 2.2 μM of compound, and the number of RVFV foci was determined 24 hours later. Antiviral efficacy was calculated as an estimated 50% effective concentration ($EC_{50}$) from the number of RVFV foci detected in comparison to vehicle control (S1 Table). Following detection of the RVFV foci, cytotoxicity concentrations ($CC_{50}$) were estimated by qualitatively assessing monolayer integrity by staining the cell monolayers with crystal violet and measuring the optical density of crystal violet staining, which is proportional to the number of viable cells. Screening hits were defined as compounds that i) had an estimated $EC_{50}$ determined from the four-point screening assay of <20 μM and ii) by measuring monolayer integrity using crystal violet, as commonly done for cell-based bioassays.

Forty-seven screening hits were identified (S1 Table). Thirty-nine of 174 troponoids screened (22%) were hits, with 34 of them being αHTs, three being TRPs, and two being TTPs. In contrast, only eight of 223 non-troponoids (3.6%) were hits. Seven of these eight hits were HPDs. The hit rate among the 24 HPDs screened (29%) was similar to that of the αHTs. The remaining screening hit was a DHN.

## 3.3. Quantification of antiviral activity against RVFV

Based upon the activity of the primary screen of αHTs and HPDs and because their antiviral activity against RVFV is novel, we defined the dose-response curve for 25 αHTs, 4 HPDs, and 6 additional compounds selected to broaden the chemical diversity. This also served to spot-check compounds with poor estimated $EC_{50}$ values. In these studies, Vero cells were treated with 8 different concentrations ranging from 60 to 0.024 μM at the time of infection and their ability to prevent virus replication was measured by comparing the number of viral infection foci to wells treated with vehicle control. $EC_{50}$ ranged from 1.2 to 40.8 μM (Table 1), with 21 of the top 30 compounds being αHTs. Interestingly, two closely related compounds AG-II-18-P (308), a thiophene substituted αHT and the closely related furan counterpart AG-I-183-P (309), had an $EC_{50}$ of 1.2 μM and 1.6 μM respectively, and were the two best compounds identified.

To better understand potential cellular cytotoxicity in the context of infection, crystal violet staining of the cell monolayers was quantified by absorbance and $CC_{50}$ values were calculated after foci were counted. The $CC_{50}$ ranged from 44 to >120 μM in Vero cells, with 24 compounds having a $CC_{50}$ of >120 μM. Second, cytotoxicity was assessed in the HepG2 derivative HepDES19 [46] to model longer-duration compound exposure in hepatocytes. Cells were

**Table 1. Top antiviral hits.**

| Compound number | Name [1] | Chemotype [2] | EC$_{50}$ (µM) | CC$_{50}$ (µM) [3] | SI | Chemist |
|---|---|---|---|---|---|---|
| α-Hydroxytropolones | | | | | | |
| 308 | AG-II-18-P | αHT | 1.2 | 111 | 92 | Murelli |
| 309 | AG-I-183-P | αHT | 1.6 | >120 | 75 | Murelli |
| 362 | DS-I-69 | αHT | 2.5 | 80 | 32 | Murelli |
| 694 | NBA-I-127 Bis | αHT | 3.7 | >120 | 32 | Murelli |
| 359 | AG-II-108-C | αHT | 5.1 | >120 | 24 | Murelli |
| 696 | NBA-I-128 Bis | αHT | 6.0 | 118 | 20 | Murelli |
| 1017 | AL-23 | αHT | 8.0 | >120 | 15 | Murelli |
| 867 | DS-1-124 | αHT | 8.7 | >120 | 14 | Murelli |
| 702 | NBA-I-159 Mono | αHT | 8.8 | >120 | 14 | Murelli |
| 1039 | AB-3-45 | αHT | 8.9 | >120 | 13 | Murelli |
| 336 | YA-I-78 | αHT | 9.0 | 70 | 8 | Murelli |
| 330 | NBA-I-14 | αHT | 10.4 | >120 | 12 | Murelli |
| 698 | NBA-I-150 | αHT | 11.7 | >120 | 10 | Murelli |
| 311 | AG-II-3-P | αHT | 11.8 | 115 | 10 | Murelli |
| 210 | MolMoll 19617 | αHT | 11.8 | >120 | 10 | Purchased |
| 390 | AB-2-70 | αHT | 11.9 | >120 | 10 | Murelli |
| 838 | NBA-I-130 | αHT | 12.7 | >120 | 9 | Murelli |
| 704 | NBA-I-160 | αHT | 14.3 | >120 | 8 | Murelli |
| 331 | NBA-I-31 | αHT | 15.8 | 80.4 | 5 | Murelli |
| 703 | NBA-I-159 Bis | αHT | 16.1 | >120 | 7 | Murelli |
| 539 | AB-2-91 | αHT | 18.6 | 71 | 4 | Murelli |
| 840 | NBA-I-155-Mono | αHT | 31.6 | >120 | 4 | Murelli |
| 320 | NBA-I-13 | αHT | 34.0 | >120 | 4 | Murelli |
| 335 | DH-2-60 | αHT | 40.8 | >120 | 3 | Murelli |
| Thiotropolones | | | | | | |
| 680 | BE1105 | TTP | 6.3 | >120 | 19 | Elgendy |
| 686 | BE1111 | TTP | 6.2 | >120 | 19 | Elgendy |
| Tropolones | | | | | | |
| 341 | Specs AP-355/40802214 | TRP | 25.2 | >120 | 5 | Purchased |
| 340 | Specs AP-355/40633884 | TRP | 27.3 | 70 | 4 | Purchased |
| 342 | Specs AP-355/40633885 | TRP | 55.5 | >120 | 2 | Purchased |
| N-Hydroxypyridinediones | | | | | | |
| 670 | ZEV-V7 | HPD | 14.0 | >120 | 9 | Zoidis |
| 668 | ZEV-V5 | HPD | 19.2 | >120 | 6 | Zoidis |
| 518 | ZEV-V3 | HPD | 19.5 | >120 | 6 | Zoidis |
| 515 | ZEV-E2 | HPD | 24.3 | >120 | 5 | Zoidis |
| Dihydronapthalene | | | | | | |
| 327 | Aldrich Select CNC_ID 444085867 | DHN | 39.7 | 44 | 1.1 | Purchased |
| Nucleoside Analogue | | | | | | |
| EIDD-1931 | β-D-N4-HydroxycytidineN4-Hydroxycytidine | Nuc | 5.2 | >120 | 23 | Purchased |
| | Ribavirin | Nuc | 22.0 | >120 | 5 | Purchased |

[1] Chemist's name, common name, or vendor catalog number.

[2] αHT, α-Hydroxytropolone; TRP, tropolone; TTP, thiotropolone; DHN, dihydronapthalene; HPD, N-Hydroxypyridinedione; FLV, flavenoid; DOB, dioxobutanoic acid; HXT, hydroxyxanthanone; TPD, thieopyrimidinone; ACT, aminocyanothiophene.

[3] Values of 120 indicate the data were at or above the upper limit of quantification in the assay.

**Table 2. EC$_{50}$ against RVFV replication.**

| Compound number | Compound name | Chemotype | A549 | | | Vero | | |
|---|---|---|---|---|---|---|---|---|
| | | | EC$_{50}$, μM | CC$_{50}$ μM | SI | EC$_{50}$, μM | CC$_{50}$ μM | SI |
| 309 | AG-I-183-P | αHT | 5.9 | 98.1 | 16.8 | 1.6 | >120 | 75.0 |
| 686 | BE1111 | TTP | 7.9 | 67.6 | 8.6 | 6.3 | >120 | 19 |
| 308 | AG-II-18-P | αHT | 8.4 | 119.0 | 14.2 | 1.2 | >120 | 100.0 |
| 390 | AB-2-70 | αHT | 9.4 | 224.2 | 23.9 | 11.9 | >120 | 10.1 |
| 680 | BE1105 | TTP | 11.6 | >240 | 20.7 | 6.3 | >120 | 19.0 |
| 670 | ZEV-V7 | HPD | 15.3 | >240 | 15.7 | 14 | >120 | 8.6 |
| 840 | NBA-I-155-Mono | αHT | 15.4 | 141.6 | 9.2 | 31.6 | >120 | 3.8 |
| 1039 | AB-3-45 | αHT | 15.5 | >240 | 15.5 | 8.9 | >120 | 13.5 |
| 518 | ZEV-V3 | HPD | 17.2 | 16.5 | 1.0 | 20.0 | >120 | 6.0 |
| 331 | NBA-I-31 | αHT | 17.7 | >240 | 13.6 | 15.8 | 80.4 | 5.0 |
| 327 | Aldrich Select CNC_ID 444085867 | DHN | 19.6 | 27.5 | 1.4 | 39.7 | 42.9 | 1.1 |
| 704 | NBA-I-160 | αHT | 20.0 | 43.0 | 2.2 | 14.3 | >120 | 8.4 |
| 867 | DS-1-124 | αHT | 33.0 | >240 | 7.3 | 8.7 | >120 | 13.8 |
| 668 | ZEV-V5 | HPD | 35.7 | 82.0 | 2.3 | 19.2 | >120 | 6.3 |
| 330 | NBA-I-14 | αHT | 42.6 | >240 | 5.6 | 10.4 | >120 | 11.5 |
| 320 | NBA-I-13 | αHT | 90.3 | 154.1 | 1.7 | 33.96 | >120 | 3.5 |
| 517 | ZEV-V2 | HPD | 95.8 | 71.8 | 0.7 | >120 | >120 | - |
| 515 | ZEV-E2 | HPD | >120 | 93.6 | 0.6 | 24.3 | >120 | 4.9 |
| 335 | DH-2-60 | αHT | >120 | 96.2 | 0.3 | 40.8 | >120 | 2.9 |

treated with a range of compound concentrations for three days and mitochondrial function was measured using MTS assays. CC$_{50}$ values ranged 1.8 to >100 μM in the hepatoblastoma cells (S2 Table).

To validate compound efficacy and cytotoxicity, we completed additional dose response curve experiments in A549 cells (Table 2 and Fig 2). We selected A549 cells, a human alveolar basal epithelial cell line, for two reasons, the potential for respiratory exposure of humans via aerosol or droplets and the epithelial nature of A549s because of the evidence that natural infection of RVFV leads to infection of epithelial cells within the kidney, liver, and spleen [5, 56, 57]. There was a concordance for the majority of compounds between the EC$_{50}$ value defined in Vero cells and the EC$_{50}$ concentration defined in A549 cells. The values from the corresponding compounds in Vero cells is provided next to the data for A549s. In A549 cells, both compound AG-II-18-P (308), a thiophene substituted αHT, and its furan counterpart (309) had the lowest EC$_{50}$ values of 8.4 and 5.9 μM, respectively. To quantify cytotoxicity independent of viral infection, cellular toxicity was quantified by measuring intracellular protease release. In this manner, we were able to quantify cytotoxicity in the same cell lines which were used to determine efficacy. In these assays, A549 cells were cultured and plated as for the efficacy assays, and cells were incubated with compound for two days and cell viability measured. For the αHTs, CC$_{50}$ values ranged from 43 to >240 μM, and they ranged from 16.5 to > 240 μM for the HPDs.

## 3.4. Efficacy of compounds against wild type RVFV and LACV

In order to determine efficacy for wild type isolates of RVFV, we measured the efficacy of the αHT AG-II-18-P (308) and the nucleoside analogues NHC and ribavirin as a positive control against the highly pathogenic strain ZH501 in Vero cells (Fig 3A). Both NHC and ribavirin are

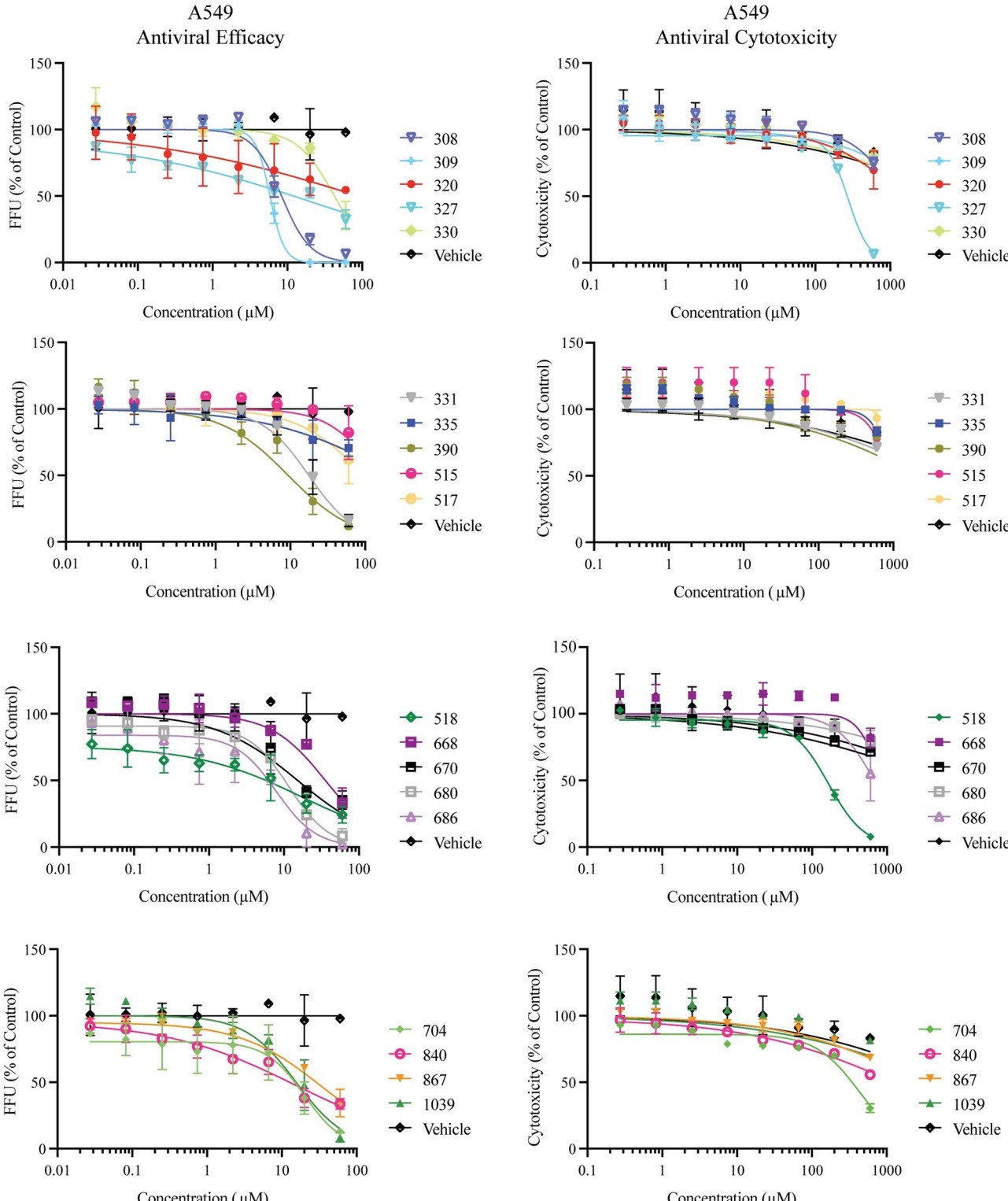

**Fig 2. *In vitro* dose-response and cytotoxicity of compounds against RVFV (MP12).** A549 cells were infected with RVFV MP12 then treated with decreasing concentrations of compound. The reduction in virus concentration was measured by FFA at twenty-four hours post infection. Data is representative of three individual experiments with two biological replicates. Error bars represent standard deviation.

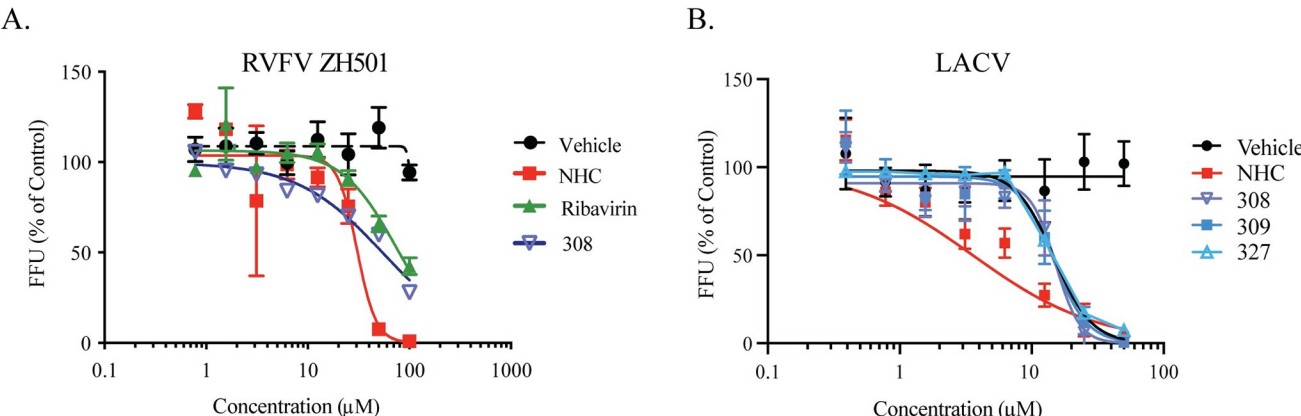

**Fig 3. Antiviral effect of compounds on bunyavirus replication.** Vero cells were infected with either RVFV ZH501 (A) or LACV (B) then treated with decreasing concentrations of antiviral compound. Viral growth was measured by FFA. Data represents three independent experiments completed with biological replicates. Error bars represent standard deviation.

small molecule inhibitors, with ribavirin having demonstrated activity against RVFV and NHC a nucleoside analog with activity against a broad array of viruses. The assay design is identical to Fig 1, with the exception that foci were measured at 18 hours post infection because of the increased rate of replication. In this assay, compound 308 had an $EC_{50}$ of 54 μM, while NHC and ribavirin had $EC_{50}$ values of 30.5 and 73.6 μM. Prior investigations of ribavirin restriction of RVFV has demonstrated similar results in Vero cells, showing the validity of the assay, and acting as a reference point for the novel compounds identified [53].

To determine if the metal chelating compounds including the αHTs AG-II-18-P (308), AG-I-183-P (309) and 16 related compounds were specific to RVFV or would have a broader activity, we investigated efficacy of these compounds against the bunyavirus LACV, as well as an unrelated flavivirus, ZIKV in Vero E6 cells. All three viral pathogens require viral metalloproteases for viral replication, while RVFV and LACV also encode a viral endonuclease used for cap-snatching [10–12, 58]. We observed a dose-dependent decrease in viral titers in the antiviral efficacy assays, with 6 of the 18 metal chelating compounds against LACV virus resulting in $EC_{50}$'s ranging between 12.7 and 89.15 μM, while the nucleoside analogue β-D-N4-Hydroxycytidine had an $EC_{50}$ of 3.6 μM (Fig 3B and Table 3). Of the 6 compounds that had activity against LACV, only 2 had activity against ZIKV. The αHT compound 704 and the TTP compound 680 had $EC_{50}$'s of 23.0 and 19.5 μM respectively, with the nucleoside analogue β-D-N4-Hydroxycytidine having a $EC_{50}$ of 1.7 μM. The ability of a subset of compounds to inhibit the growth and replication of both RVFV and LACV speaks to the potential of these metal chelating compounds to be developed as potential bunyaviridae therapeutics, or potentially as phlebovirus (RVFV)-or orthobunyavirus (LACV)-specific antivirals.

### 3.5. Selective indexes

Selectivity indexes (SI or $CC_{50}/EC_{50}$) were calculated for all compounds with $EC_{50}$ values based on cytotoxicity in both Vero cells after one day compound treatment and in A549 cells after two days of treatment. SIs ranged from <1 to 402 in Vero cells (Table 1 and S2 Table) and <1 to 23.9 in A549 cells (Table 2). Confirmed hits were defined as compounds with SIs > 5 in Vero cells (n = 34) because this indicates that the reduction in RVFV foci was not a result of non-specific killing of the Vero cells in which the assays were completed (Table 1).

**Table 3. EC$_{50}$ against LACV and ZIKV replication.**

| Compound number | Compound name | Chemotype | LACV EC$_{50}$, µM | ZIKV EC$_{50}$, µM |
|---|---|---|---|---|
| 308 | AG-II-18-P | αHT | 15.1 | >120 |
| 309 | AG-I-183-P | αHT | 14.6 | >120 |
| 320 | NBA-I-13 | αHT | >120 | >120 |
| 327 | Aldrich Select CNC_ID 444085867 | DHN | 12.7 | >120 |
| 330 | NBA-I-14 | αHT | >120 | 32.7 |
| 335 | DH-2-60 | αHT | >120 | >120 |
| 390 | AB-2-70 | αHT | >120 | >120 |
| 515 | ZEV-E2 | HPD | >120 | >120 |
| 517 | ZEV-V2 | HPD | >120 | >120 |
| 518 | ZEV-V3 | HPD | >120 | >120 |
| 668 | ZEV-V5 | HPD | >120 | >120 |
| 670 | ZEV-V7 | HPD | >120 | >120 |
| 680 | BE1105 | TTP | 31.3 | 23 |
| 686 | BE1111 | TTP | 45.4 | >120 |
| 704 | NBA-I-160 | αHT | 89.15 | 19.5 |
| 840 | NBA-I-155-Mono | αHT | >120 | >120 |
| 867 | DS-1-124 | αHT | >120 | >120 |
| 1039 | AB-3-45 | αHT | >120 | >120 |
| NHC | β-D-N4-Hydroxycytidine | NUC | 3.6 | 1.74 |

Structures of all compounds in Tables 1 and 2 are in S2 Data, and structures of all confirmed hits are in S3 Data.

## 4. Discussion

Most primary hit compounds against RVFV were either αHTs or HPDs, but TRP, TTP, and DHN hits were also found. This distribution of hits is partially due to sampling bias owing to the disproportionate number of αHTs in the compound collection screened. Sampling bias, however, does not fully explain the hit distribution because only a minority of the αHTs screened were active, and because there were many chemotypes in the compound collection where hits were not found. These included dioxobutanoic acids, hydroxyxanthanones, thieno-pyrimidinones, pyridinepiperazinthieonpyrimidins, N-biphenyltrihydroxybenzamides, and aminocyanothiophenes. EC$_{50}$ values of the 62 compounds for which quantitative data were obtained ranged from 0.1 to >120 µM (S1 and S2 Tables). Selective indexes for these compounds in Vero cells in which the screening was conducted ranged from 1.1 to 1200. Twenty-eight compounds were confirmed hits based upon their selective index (SIs > 5 in Vero cells, 7.5% of the compounds screened) indicating that they were due to bona fide inhibition of RVFV rather than secondary effects of cytotoxicity. However, the increased CC$_{50}$ values in the A549 (16.5 to >240 µM) and HepDES19 cells (<1 to 48 µM) indicate that compounds active against RVFV replication can have cytotoxicity in human lung and liver-derived cells, which must be addressed during subsequent hit-to-lead medicinal chemistry campaigns. One avenue for reducing cytotoxicity, at least for the αHTs, may be to reduce the lipophilicity and number of aromatic rings in the molecules because these parameters correlate with αHT toxicity [38]. RVFV infections proceed rapidly *in vivo*, so optimizing these hits to achieve toxicity profiles suitable for a one- to two-week treatment regimen in RVFV-infected patients or animals will likely be enough to yield usable drugs.

Identification of primary screen hits among the αHT, HPD, TRP, TTP, and DHN chemotypes indicates that a range of compounds can inhibit RVFV replication, but the lack of hits among the other chemotypes screened implies that there is specificity to RVFV inhibition (Fig 4). This is further supported by the efficacy of a subset of αHT class of compounds against LACV, with two of the compounds (308, 309) having activity against LACV, but not the unrelated flavivirus, Zika virus. The potential for specific inhibition of RVFV was confirmed by the wide range of inhibition patterns observed during counter-screening (S2 Table). For example, compound 362 had an $EC_{50}$ of 2.6 μM vs. RVFV, an $IC_{50}$ against human ribonuclease H1 of 212 μM, was inactive against *E. coli* growth, and was only modestly effective against the pathogenic fungus *C. neoformans* (minimum 80% inhibitory concentration ($MIC_{80}$) = 24 μM). This indicates that although these chemotypes which can bid to the active sites of metalloenzymes and can have broad anti-microbial activity, individual compounds can be selective through specific interactions with their various targets (reviewed in [59]).

The 174 troponoids (αHTs, TRPs, TTPs) evaluated yielded 28 confirmed hits with TI values ≥ 5 (Table 1). Inhibition of RFVF by tropolones appears to require an intact metal ion chelating trident on the compound, which implies an interaction with two closely-spaced divalent cations on the target molecule due to the compounds' known mechanisms of metal chelation (Fig 4a) [59]. For example, while the tropolones β- and γ-thujaplicin were inactive against the virus at concentrations upwards of 120 μM, the αHT β–thujaplicinol (46) was active, with an $EC_{50}$ of 13.8 μM (Fig 2A). These trends extended to additional αHTs, of which 11 different molecules had $EC_{50}$ values under 10 μM (Table 1), the most potent of which had an $EC_{50}$ of 1.2 μM (308), (S3 Data). These more potent molecules had a broad range of appendages (Fig 4B), such as ketone (308, 309, 359, 362), amide (867, 1017, 1039), mono- (702) and bis (694, 696) thioethers, sulfoxide (336), and included a 3,7-dihydroxytropolone (362). Thus, there is tolerance to a variety of functional groups (Fig 4c). Apart from the αHTs, four additional tropolones had measurable $EC_{50}$ values (TRPs 340, 341, 342 and 359, $EC_{50}$ = 5.1–55.5 μM), each of which had a carbonyl appendage α to the tropolone oxygens that could provide an alternative third cation contact point. However, only 341 had an $EC_{50}$ under 10 mM (S2 Data). Intriguingly, both acylated thiotropolones (680, 686) had sub-mM activity despite the lack of any tridentate cation binding motif. It seems possible that these thioester linkages could undergo cleavage in the cell, and that the active component in both instances is the free thiotropolone, as has been postulated previously for their potent anti-*Cryptococcus neoformans* activity [37].

Seven HPD primary hits were found among the 24 HPDs screened. Four of these were confirmed hits, 515, 518, 668, and 670, with $EC_{50}$ values ranging from 14.0–24.3 μM (Fig 2). This is insufficient to generate a meaningful structure-activity relationship, but trends can be inferred. The oxygen trident of the HPD scaffold is essential for its activity, as the loss of any one of these oxygens results in inactive compounds. In each case it is presumed based on data with other HPDs against HBV [47] that strong ionic interactions, along with charge-assisted hydrogen bonds potentially anchor the chelator moiety of HPDs (N-Hydroxyimide group) to a cellular or viral metalloenzyme ensemble comprised of the two positively charged $Mg^{++}$ ions. Lastly, aromatic substitutions at the imine nitrogen are tolerated that carry modifications including halogen electron withdrawing groups and an alkyl electron donating group.

The mechanism(s) of action of these RVFV inhibitors are unknown and could involve inhibition of viral and/or cellular proteins. However, inhibiting one or more mono- or di-metalloenzymes needed for viral replication by chelating their active site cations is a potential mechanism. The rationale for implicating metal chelation comes from the compound structures and their known activities against HIV and HBV [16–18]. The αHTs and HPDs have metal chelating tridents suitable for binding to the $Mg^{++}$ ions in di-metalloenzyme active sites,

**Fig 4. Representative structures of RFVF inhibitors.** (A) Inactive and active troponoid natural products, illustrating preference for oxygen triad, along with common nuclease inhibition mode for αHTs. (B) Synthetic αHTs with activity under 10 μM against RVVF, demonstrating broad substitution tolerance. (C) Representative examples of alternative scaffolds with activity against RFVF.

and the αHTs are known to work by this mechanism against the HIV ribonuclease H and/or integrase [17, 21]. However, the failure of many compounds with known ability to inhibit divalent cation containing metalloenzymes (~70% of the αHTs did not inhibit RVFV growth) indicates that metal chelation by itself is insufficient, presumably because additional compound: target interactions are needed to provide sufficient binding affinity to inhibit viral replication. One potential target for these inhibitors is the RVFV L protein, where either the viral RNA-dependent RNA polymerase (RdRp) activity or the cap-snatching endonuclease activity of the

viral L protein could be affected [32, 60]. This is because both are di-metalloenzymes that catalyze a reaction required for viral replication.

## 5. Conclusions

Screening for RVFV replication inhibitors among compounds selected for their similarity to inhibitors of viral nucleases identified 47 novel RVFV inhibitors. The frequent efficacy of the αHT and HPD compounds screened against RVFV replication indicates that these two scaffolds are promising candidates for optimization into anti-RVFV drugs that target metalloenzymes for use against human and/or veterinary infections. Cytotoxicity was observed in human hepatoblastoma cells, indicating that identifying and mitigating the causes of cytotoxicity will be key to optimizing these hits. The conserved features of viral replication among *Bunyavirales*, and the activity of these compounds against RVFV and a subset of these compounds having activity against LACV implies that these hits hold potential for development into treatments for related pathogens, including Hantaan virus, severe fever with thrombocytopenia syndrome virus, and Crimean-Congo hemorrhagic fever virus.

## Supporting information

**S1 Table. Primary screening data.**
(XLSX)

**S2 Table. EC$_{50}$ value against RVFV.**
(XLSX)

**S1 Data. Experimental methods for compound generation and validation of compounds.**
(PDF)

**S2 Data. Structures of all compounds in Tables 1 and 2.**
(PDF)

**S3 Data. Structures of all confirmed hit compounds.**
(PDF)

## Acknowledgments

We thank Qilan Li, Brienna Milleson, Alaina Knier, and Elena Lomonosova for technical assistance. We thank Drs. Mark Campbell and Christopher Eickoff as SROs for the maintenance Saint Louis University select agent program.

## Author Contributions

**Conceptualization:** John E. Tavis, Amelia K. Pinto, James D. Brien.

**Data curation:** E. Taylor Stone, John E. Tavis, James D. Brien.

**Formal analysis:** Elizabeth Geerling, Maria C. Mai, Mariah Hassert, Feng Cao, Maureen J. Donlin, Bahaa Elgendy, Grigoris Zoidis, James D. Brien.

**Funding acquisition:** Grigoris Zoidis, John E. Tavis, Amelia K. Pinto, James D. Brien.

**Investigation:** Elizabeth Geerling, Valerie Murphy, Maria C. Mai, E. Taylor Stone, Andreu Gazquez Casals, Mariah Hassert, Austin T. O'Dea, Feng Cao, Maureen J. Donlin, Mohamed Elagawany, Bahaa Elgendy, Vasiliki Pardali, Erofili Giannakopoulou, Grigoris Zoidis,

Daniel V. Schiavone, Alex J. Berkowitz, Nana B. Agyemang, Ryan P. Murelli, John E. Tavis, Amelia K. Pinto, James D. Brien.

**Methodology:** Elizabeth Geerling, Valerie Murphy, E. Taylor Stone, Andreu Gazquez Casals, Mariah Hassert, Austin T. O'Dea, Feng Cao, Maureen J. Donlin, Bahaa Elgendy, Vasiliki Pardali, Erofili Giannakopoulou, Grigoris Zoidis, Daniel V. Schiavone, Alex J. Berkowitz, Ryan P. Murelli, John E. Tavis, Amelia K. Pinto.

**Project administration:** Elizabeth Geerling, James D. Brien.

**Resources:** Maureen J. Donlin, Ryan P. Murelli.

**Supervision:** Ryan P. Murelli, John E. Tavis, Amelia K. Pinto, James D. Brien.

**Writing – original draft:** John E. Tavis, James D. Brien.

**Writing – review & editing:** Elizabeth Geerling, Maria C. Mai, Grigoris Zoidis, Ryan P. Murelli, John E. Tavis, Amelia K. Pinto, James D. Brien.

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
