## [Decision Letter · Decision Letter 0]

3 Jun 2022

PONE-D-22-12188Metal coordinating inhibitors of Rift Valley fever virus replicationPLOS ONE

Dear Dr. Brien,

Thank you for submitting your manuscript to PLOS ONE. After careful consideration, we feel that it has merit but does not fully meet PLOS ONE’s publication criteria as it currently stands. Therefore, we invite you to submit a revised version of the manuscript that addresses the points raised during the review process.

Please pay careful attention to the specific comments of both reviewers. 

The following items must be addressed:

The authors should perform some experiments to test the hypothesis that the antiviral compounds identified may target the nuclease/ cap-snatching activity of the L protein. 

A figure describing the SAR should be added. 

Please also address the following comment from Reviewer #2 "For multiple compounds the dose-response curves are not complete. Listed EC50 values might not be accurate."

We look forward to receiving your revised manuscript.

Kind regards,

Kylene Kehn-Hall

Academic Editor

PLOS ONE

Journal Requirements:

"I have read the journal's policy and the authors of this manuscript have the following competing interests: pending patent application.

AP, GZ, JB, JT, and RM are inventors on a pending US patent application covering use of these compounds to treat Bunyavirus infections."

Reviewers' comments:

Reviewer's Responses to Questions

**Comments to the Author**

1. Is the manuscript technically sound, and do the data support the conclusions?

Reviewer #1: Partly

Reviewer #2: Yes

2. Has the statistical analysis been performed appropriately and rigorously? 

Reviewer #1: Yes

Reviewer #2: Yes

3. Have the authors made all data underlying the findings in their manuscript fully available?

Reviewer #1: Yes

Reviewer #2: Yes

4. Is the manuscript presented in an intelligible fashion and written in standard English?

Reviewer #1: Yes

Reviewer #2: Yes

5. Review Comments to the Author

Reviewer #1: Geerling et al. have exploited the fact that divalent cations play essential roles in catalysis and/or substrate binding of many enzymes, including those encoded by viruses. The authors cite notable examples of the use of small molecules to chelate active site divalent cations to inhibit metalloenzymes, including numerous FDA approved drugs. The authors have used these observations as the basis for screening known chelator molecules and structurally similar molecules for inhibitory activity against a potent viral pathogen, RVFV.

Thus, in the primary screen 375 (line 101) or 397 (line 264) compounds with known or plausible metal chelation ability were tested in a live-cell assay for the ability to reduce focus forming units (visualized by staining with antibody against the RVFV nucleoprotein) after 24 hours. Cytotoxicity was estimated by subsequent staining of the treated cells with crystal violet and visual inspection of the integrity of the cell monolayers in the cell culture wells.

The compounds with the highest potency and least cytotoxicity were retested in dose response curves in liver cells (HEP DES19) for cytotoxicity and lung cells (A549) for antiviral activity. Cross-testing for antiviral activity against a highly pathogenic strain of RVFV and against LACV from another family of Bunyaviruses was used to validate the initial hits.

Overall the results of the screening are interesting and give some optimism for further development of these compounds toward antiviral drugs. The primary screening method using low MOI infection of VeroE6 cells, methylcellulose overlay followed by immunostaining looks solid. Follow up of best compounds using different cell types and viruses is good but could be expanded slightly to support mechanistic assertions.

The only obvious weakness of the manuscript as presented is in the lack of mechanistic analysis. Although this might not be essential for an initial screening study, it is a little puzzling that there is rather extensive discussion of a specific chelation mechanism, and that the nuclease/ cap-snatching activity of the L protein may be the target. It would certainly be significant to identify inhibitors of cap-snatching activity for the bunyaviruses. However, there are no data to support these assertions.

One relatively easy experiment to preliminarily test this hypothesis would be to assay antiviral activity of select compounds against an RNA virus that does not employ cap-snatching (in addition to the demonstration of activity against LACV). If activity is lost, this is one data point that supports the hypothesis. If activity is retained, then the hypothesis (and discussion) would require some modification.

The authors use the observation that some of the compounds display activity against LACV as well as RVFV as an indication of specificity, but orthobunyaviruses and phleboviruses are fairly divergent. Although there may not be structural models available, it could be useful to indicate a percent amino acid conservation of the L proteins between these two families, or some other indication that the divalent ion binding sites of these two enzymes are similar.

The authors do a pretty good job of informally describing the potential SARs of identified a-HTs and HPDs, but a figure summarizing the gain/loss of activity corresponding to the changes to the scaffold would be helpful.

Minor point:

The explanation for why A549 cells were used in the dose response assays is a little curious, since it is noted that RVFV pathology is most pronounced in kidney, liver, spleen, not lung.

Reviewer #2: Geerling and colleagues describe the screening of metal coordinating antiviral compounds for Rift Valley fever virus (RVFV) and La Crosse (LACV) bunyavirus. Compounds were selected based on previous data for these compound classes to be effective for other viruses, such as HIV, HBV, and influenza. Currently, no approved therapeutics exist for RVFV and LACV and there is need for identification and development.

The authors developed focus forming assays for RVFV and LACV to perform antiviral screens in 96 well format. Compounds with antiviral activity were further characterized in dose-response studies against both the vaccine strain MP-12 for RVFV and the pathogenic RVFV ZH501 isolate. Polymerase inhibitors were identified with activities in the low to mid uM range. Identified compounds belong to the classes of alpha-Hydroxytropolones and N-Hydroxypyridinediones.

Overall, this is a fairly straight forward study, however, with limited compound characterization. Future studies will be necessary to further characterize individual compounds. The following comments should be addressed to improve the manuscript:

The abstract refers to 47 identified compounds and the Conclusion to 27.

Line 55: Please add references.

Lines 67 and 70: The same information about the lack of therapeutics is provided.

Lines 75 and 81: Repeat information is listed.

Line 91: Please add references for the listed antiviral compounds.

Line 92: Are more recent data for approved drugs available (after 2017)?

Line 133: Please add the institution for Drs. Buller and Hise.

Line 179: The information for LACV antibodies has already been described in paragraph 2.3

Compound cytotoxicity: Why did the authors not determine CC50 values in un-infected Vero cells?

Lines 239-243: Please add NHC to the figure legend

Line 251: If a reference is available for NHC testing against RVFV, please add.

Lines 307-309: The authors should add that aerosol transmission of RVFV has been described, which would justify the use of A549 cells.

Lines 309-311: While there is concordance in EC50 values for multiple compounds in both cells lines tested, there are also multiple compounds with differences (e.g., 308, 327, 867, 330, 515).

Why did the authors only test compound 308 against ZH501 and not 309 as well? Both compounds have the highest SI values (and 309 was tested against LACV).

Figure 1: The legend in B can be removed, since the same information is listed on the x-axis.

Figure 2: For multiple compounds the dose-response curves are not complete. Listed EC50 values might not be accurate.

6. PLOS authors have the option to publish the peer review history of their article (what does this mean?). If published, this will include your full peer review and any attached files.

Reviewer #1: No

Reviewer #2: No

---

## [Author Response · Author response to Decision Letter 0]

19 Aug 2022

We have attached a document which has all of the responses to the reviewers comments, that maybe easier to read then the information below. 

The information below is the exact text as in the attached Response to Reviewers document.

Response to Reviewer’s Comments

Review Comments to the Author

Reviewer #1: Geerling et al. have exploited the fact that divalent cations play essential roles in catalysis and/or substrate binding of many enzymes, including those encoded by viruses. The authors cite notable examples of the use of small molecules to chelate active site divalent cations to inhibit metalloenzymes, including numerous FDA approved drugs. The authors have used these observations as the basis for screening known chelator molecules and structurally similar molecules for inhibitory activity against a potent viral pathogen, RVFV.

Thus, in the primary screen 375 (line 101) or 397 (line 264) compounds with known or plausible metal chelation ability were tested in a live-cell assay for the ability to reduce focus forming units (visualized by staining with antibody against the RVFV nucleoprotein) after 24 hours. Cytotoxicity was estimated by subsequent staining of the treated cells with crystal violet and visual inspection of the integrity of the cell monolayers in the cell culture wells.

The compounds with the highest potency and least cytotoxicity were retested in dose response curves in liver cells (HEP DES19) for cytotoxicity and lung cells (A549) for antiviral activity. Cross-testing for antiviral activity against a highly pathogenic strain of RVFV and against LACV from another family of Bunyaviruses was used to validate the initial hits.

Overall the results of the screening are interesting and give some optimism for further development of these compounds toward antiviral drugs. The primary screening method using low MOI infection of VeroE6 cells, methylcellulose overlay followed by immunostaining looks solid. Follow up of best compounds using different cell types and viruses is good but could be expanded slightly to support mechanistic assertions.

The only obvious weakness of the manuscript as presented is in the lack of mechanistic analysis. Although this might not be essential for an initial screening study, it is a little puzzling that there is rather extensive discussion of a specific chelation mechanism, and that the nuclease/ cap-snatching activity of the L protein may be the target. It would certainly be significant to identify inhibitors of cap-snatching activity for the bunyaviruses. However, there are no data to support these assertions.

We appreciate the reviewer’s comments and have eliminated the discussion of a specific chelation mechanisms within the results and discussion sections of the manuscript.

One relatively easy experiment to preliminarily test this hypothesis would be to assay antiviral activity of select compounds against an RNA virus that does not employ cap-snatching (in addition to the demonstration of activity against LACV). If activity is lost, this is one data point that supports the hypothesis. If activity is retained, then the hypothesis (and discussion) would require some modification.

We have investigated a panel of 18 compounds that were effective against RVFV, against Zika virus and LACV and quantified compound efficacy. Of the 18 compounds, 6 compounds that had activity against RVFV also had activity against LACV, with only two of those 6 compounds having activity against ZIKV.

The authors use the observation that some of the compounds display activity against LACV as well as RVFV as an indication of specificity, but orthobunyaviruses and phleboviruses are fairly divergent. Although there may not be structural models available, it could be useful to indicate a percent amino acid conservation of the L proteins between these two families, or some other indication that the divalent ion binding sites of these two enzymes are similar.

We agree with the reviewers that within the genus bunyaviridae that orthobunyaviruses such as LACV and phleboviruses such as RVFV have divergent amino acid sequences. However, there are the number of 

structural commonalities between the L proteins of both viral families. The enzymatic domains of the L proteins of both are co-linear with the endonuclease domain at the 5’ end of the protein and the cap binding domain at the 3’ end of the protein (1-3), while the four tunnels which lead to the catalytic center are structurally similar in size (2, 3). Specifically, the 5’ endonuclease for both LACV and RVFV are classified as His+, and contain PD(E/D)K motif (reviewed in (4)). The biochemical and structural similarities of IAV endonucleases and LACV/RVFV endonuclease has led to the proposal that IAV endonuclease inhibitors could be used against the viruses as well.

The authors do a pretty good job of informally describing the potential SARs of identified a-HTs and HPDs, but a figure summarizing the gain/loss of activity corresponding to the changes to the scaffold would be helpful.

We have added figure four and text describing the SAR of the αHTs and HPDs. This will allow the readers to understand the gain and loss of activity corresponding to the changes to the compounds scaffold.

Minor point:

The explanation for why A549 cells were used in the dose response assays is a little curious, since it is noted that RVFV pathology is most pronounced in kidney, liver, spleen, not lung. 

We chose to use A549 cells for three key reasons: A549 are human cells commonly used in virological assays; they represent lung epithelial cells which can be a potential target of aerosol/droplet transmission of RVFV, a route which can occur during the processing of wild animals for food; and A549 cells, although derived from the lung, are epithelial cells, which is the cell type infected in the kidney, liver and spleen.

Reviewer #2: Geerling and colleagues describe the screening of metal coordinating antiviral compounds for Rift Valley fever virus (RVFV) and La Crosse (LACV) bunyavirus. Compounds were selected based on previous data for these compound classes to be effective for other viruses, such as HIV, HBV, and influenza. Currently, no approved therapeutics exist for RVFV and LACV and there is need for identification and development.

The authors developed focus forming assays for RVFV and LACV to perform antiviral screens in 96 well format. Compounds with antiviral activity were further characterized in dose-response studies against both the vaccine strain MP-12 for RVFV and the pathogenic RVFV ZH501 isolate. Polymerase inhibitors were identified with activities in the low to mid uM range. Identified compounds belong to the classes of alpha-Hydroxytropolones and N-Hydroxypyridinediones.

Overall, this is a fairly straight forward study, however, with limited compound characterization. Future studies will be necessary to further characterize individual compounds. The following comments should be addressed to improve the manuscript:

We greatly appreciate the comments and criticism provided. We have addressed every point thereby improving the text of the manuscript, including adding references or by providing a clear response with the rationale.

The abstract refers to 47 identified compounds and the Conclusion to 27.

The text within the manuscript has been clarified to describe the 47 novel inhibitors identified.

Line 55: Please add references.

References were added to support our statement.

Lines 67 and 70: The same information about the lack of therapeutics is provided.

Lines 75 and 81: Repeat information is listed.

We have removed the repeat information and have edited the text to make sure it is easy to read and understand. 

Line 91: Please add references for the listed antiviral compounds.

References for each antiviral compound were added to the manuscript.

Line 92: Are more recent data for approved drugs available (after 2017)?

This reference is a review that is highly focused on mechanisms of metalloenzymes that has not been updated since 2017, so we cannot provide a newer reference. The reference continues to support the underlying argument that metal chelation is a viable drug mechanism.

Line 133: Please add the institution for Drs. Buller and Hise.

Their respective institutions were added to the text of the manuscript.

Line 179: The information for LACV antibodies has already been described in paragraph 2.3

Compound cytotoxicity: Why did the authors not determine CC50 values in un-infected Vero cells?

In the Vero cell assays we focused on the outcome of virally infected compound treated cells to permit rigorous interpretation of the efficacy of the compounds. Vero cells are a non-human primate cell line, our experimental plan was to investigate compound toxicity in HepDES19 cells and A549 cells, two human cells lines from tissues relevant to systemic drug administration.

Lines 239-243: Please add NHC to the figure legend

We have added NHC to the figure legend.

Line 251: If a reference is available for NHC testing against RVFV, please add.

No reference is available at this time.

Lines 307-309: The authors should add that aerosol transmission of RVFV has been described, which would justify the use of A549 cells.

We agree with the reviewer that aerosol transmission to humans is one possible route of exposure, which justifies the use of A549 cells. We have added a statement and supporting literature in the text of the manuscript.

Lines 309-311: While there is concordance in EC50 values for multiple compounds in both cells lines tested, there are also multiple compounds with differences (e.g., 308, 327, 867, 330, 515).

We agree with the reviewer that the level of concordance does vary depending on the individual compound. This is not uncommon in early-stage primary screening assays as different cell lines can have different phase I and/or phase II metabolic enzymes and/or different transporters/efflux pumps. Differences in these enzymes can alter transport and stability of compounds differentially in different cell lines, causing quantitative differences in EC50 values. Such issues are routinely examined in follow up studies after primary hit compounds are identified.

Why did the authors only test compound 308 against ZH501 and not 309 as well? Both compounds have the highest SI values (and 309 was tested against LACV).

We understand and appreciate the reviewer’s question. In this case it was due to logistics. The federal permit which controls our select agent laboratory requires that ZH501 is used in isolation and requires decontamination of the room by gas sterilization. We had a small window of opportunity to evaluate compounds against the ZH501 in our select agent laboratory and compound 309 was not available at the time. 

Figure 1: The legend in B can be removed, since the same information is listed on the x-axis.

The legend has been updated accordingly.

Figure 2: For multiple compounds the dose-response curves are not complete. Listed EC50 values might not be accurate.

We agree with the reviewer’s observation that some compounds do not have a complete dose-response curve. The EC50 values are determined by non-linear curve fitting, which takes into account the upper and lower concentration limits of the compounds. Based upon the concentrations used in our assays for efficacy, any compound which did not decrease viral infection were categorized as having an EC50 of >120 μM. EC50 values above this level are biologically meaningless.

---

## [Decision Letter · Decision Letter 1]

25 Aug 2022

Metal coordinating inhibitors of Rift Valley fever virus replication

PONE-D-22-12188R1

Dear Dr. Brien,

We’re pleased to inform you that your manuscript has been judged scientifically suitable for publication and will be formally accepted for publication once it meets all outstanding technical requirements.

Kind regards,

Kylene Kehn-Hall

Academic Editor

PLOS ONE

Additional Editor Comments (optional):

Reviewers' comments:

Reviewer's Responses to Questions

**Comments to the Author**

1. If the authors have adequately addressed your comments raised in a previous round of review and you feel that this manuscript is now acceptable for publication, you may indicate that here to bypass the “Comments to the Author” section, enter your conflict of interest statement in the “Confidential to Editor” section, and submit your "Accept" recommendation.

Reviewer #1: All comments have been addressed

Reviewer #2: All comments have been addressed

2. Is the manuscript technically sound, and do the data support the conclusions?

Reviewer #1: (No Response)

Reviewer #2: Yes

3. Has the statistical analysis been performed appropriately and rigorously? 

Reviewer #1: (No Response)

Reviewer #2: Yes

4. Have the authors made all data underlying the findings in their manuscript fully available?

Reviewer #1: Yes

Reviewer #2: Yes

5. Is the manuscript presented in an intelligible fashion and written in standard English?

Reviewer #1: Yes

Reviewer #2: Yes

6. Review Comments to the Author

Reviewer #1: (No Response)

Reviewer #2: The authors have addressed the comments raised by the reviewers. The manuscript has improved and is now acceptable for publication..

7. PLOS authors have the option to publish the peer review history of their article (what does this mean?). If published, this will include your full peer review and any attached files.

Reviewer #1: No

Reviewer #2: No

---

## [Editor Report · Acceptance letter]

7 Sep 2022

PONE-D-22-12188R1 

Metal coordinating inhibitors of Rift Valley fever virus replication 

Dear Dr. Brien:

I'm pleased to inform you that your manuscript has been deemed suitable for publication in PLOS ONE. Congratulations! Your manuscript is now with our production department. 

Kind regards, 

on behalf of

Dr. Kylene Kehn-Hall 

Academic Editor

PLOS ONE